# Gender favoritism in derogatory and non-derogatory political discourse

Manuel Hons[1]*, Edgar Onea[2], Silvia Erika Kober[1]

**1** Department of Psychology, University of Graz, Graz, Austria, **2** Department of German Studies, University of Graz, Graz, Austria

* manuel.hons@uni-graz.at

## Abstract

This study investigates the interplay between Role Congruity Theory and Social Identity Theory in the context of gender perceptions among political speakers. Role Congruity Theory posits that male political speakers are perceived as more competent and likable, regardless of the recipient's gender. Conversely, Social Identity Theory suggests that individuals may exhibit a preference for speakers of the same gender due to in-group identification. We hypothesized that male participants would demonstrate a consistent preference for male speakers, driven by the synergistic effects of Role Congruity Theory and Social Identity Theory, while female participants would exhibit no such bias, resulting from the cancellation of these effects. To test this hypothesis, participants assessed German words for their offensiveness and political connotation. We employed generalized linear mixed effects models to analyze the data for two response variables – pejorative weight and political connotation – across four distinct word clusters: politically left & less derogatory, politically right & less derogatory, politically left & derogatory, and politically right & derogatory. Our findings indicate that, in terms of pejorative weight, derogatory language elicited a tendency towards in-group favoritism, which we interpret as a protective mechanism for social identity. This effect contrasts with the broader trend of male favoritism identified in other contexts. These results contribute to a deeper understanding of how gender dynamics shape the interpretation of political language, highlighting the complex interactions between social identity and speech perception in political discourse.

## Introduction

In political language, the interpretation of utterances is highly context dependent. Slurs, for example, can adopt very different meanings depending on the identity of the speaker. Croom [1] offers a comprehensive overview on differential use cases of slurs: while the word "nigger", for instance, was originally used convey hate towards black people, it has been appropriated to signal group identity, support and friendship

**Data availability statement:** The data and the analysis code can be publicly accessed at https://osf.io/sgbqf.

**Funding:** The authors declared that financial support was received for this work and/or its publication. Open access funding was provided by the University of Graz.

**Competing interests:** The authors have declared that no competing interests exist.

among African-Americans. Similarly, "bitch" and "slut" represent sexist slurs that have been appropriated for friendly in-group use. Finally, "queer" was an originally strictly pejorative term that evolved into a sexual identifier and umbrella term used by the LGBT community. Moreover, the perception of slurs is not only influenced by the group identity of the speaker [1–4], but also by the group identity of the recipient [4–6]. Carnaghy and Maass [6], for example, found that written derogatory labels for homosexuals elicited less favorable cognitive associations in heterosexual participants than they did for homosexual participants. Aside from slurs, insults and other derogatory utterances, context in the form of speaker group identity was found to generally influence how language is perceived and processed: The impact of gender [3,4,7], political attitudes [8], and accents [9] on language processing has been empirically substantiated.

Gender differences in language interpretation have been investigated extensively in many scientific fields [3,4]. However, according to Winfrey and Schnoebelen [10] our understanding of the overall role of gender differences in political communication literature is still limited. The authors provide a review on gender stereotypes regarding political speakers and conclude that gender stereotypes influence how speakers are perceived to a large degree. They argue that female political speakers are put in a paradox double-bind situation because they are expected to simultaneously demonstrate stereotypical masculine traits, as they are perceived as pivotal for strong leaders, and maintain prescriptive norms of femininity. Traits of femininity include warmth, emotional expressiveness and sensitivity, while stereotypical masculine traits comprise aggressiveness, coarseness and self-confidence [10,11]. In the attempt to satisfy this double-bind, female politicians, however, are perceived to fail in both domains, i.e., reflecting neither male nor female stereotypical qualities [12]. In this context, the reliance on gender stereotypes is most prevalent when little information is available about a speaker [13]. An analysis of the news coverage of the speeches of Canadian politicians revealed that the reports on female speakers used more aggressive and combative, and consequently stereotypically masculine, language [7] compared to reports on male counterparts. By objective measures, no noticeable differences in aggressiveness were present between male and female politicians themselves. Role congruity theory (RCT) attempts to explain the phenomenon: It argues that a disparity between societal prescriptions for women and the desired leadership traits, such as assertiveness and competitiveness, which overlap strongly with stereotypical masculine traits, exists [14,15]. Gender role incongruence leads to the judgment that women who behave according to leadership traits lack stereotypical expectations from women, while those who behave according to communal prescriptions of femininity lack what is believed to make a competent leader [14]. This ultimately causes female leaders and female potential leaders to be seen as less suited for leadership positions, but also as less likeable [15] and more aggressive [7]. In the realm of political discourse, this effect was described to be more pronounced in male than female observers [16].

Gender differences have been reported to be present in perceived political knowledge [17], political participation [18] as well as political attitudes and beliefs [19–22].

In Western, industrialized countries men tend to lean more towards conservative parties and policies, while women gravitate towards more liberal equivalents [19,21,22]. This applies even within parties [20]. More specifically, women are more supportive of increased government spending on social welfare, health and education programs as well as women and homosexual rights [22]. Women are self-reportedly more interested in gender-based topics [23] and prefer "domestic politics", such as educational and health topics, over "general" political topics, such as partisan subjects and foreign policy [24]. This is not a stable phenomenon, as it arguably arose in the 1980s [19,21,25]. Until the 1970s, a contrary pattern was observable, with women stating more conservative preferences than men [19]. Furthermore, a gender gap emerged not only in terms of the actual political attitude but also perceived political attitude: Women are perceived as more liberal, irrespective of their actual ideologies or partisan affiliations [26,27].

When dealing with socio-linguistic transmitter-receiver systems, be it in the political field or elsewhere, it is essential to discuss intergroup bias, which is characterized by two facets: in-group favoritism and out-group derogation. Social Identity Theory (SIT) [28] attempts to explain the psychosocial mechanisms surrounding in-group favoritism and out-group antagonism. The core premise of SIT is that under many social circumstances people think of themselves and others in group terms such as "us" and "them" rather than in individualistic terms. This is evidenced by a series of experiments, where the mere arbitrary categorization of participants into groups sufficed to form a social identity [29]. When a social identity is strongly embedded into one's self-concept, people are inclined to uphold a positive self-concept by defending the in-group and/or antagonizing the out-group. Further, the theory postulates strategies people can pursue to create, restore or uphold positive social identity, e.g., in case of a threat. Social creativity, for example, refers to a redefinition of a certain intergroup comparison by allocating positive rather than negative characteristics to the in-group [30,31]. One instance this cognitive mechanism is expressed by is the linguistic intergroup bias (LIB) [32] – the tendency to describe desirable in-group and non-desirable out-group behaviors in an abstract manner, and non-desirable in-group and desirable out-group behavior in a more concrete way [33,34]. The former thereby attributes actions temporal stability, implying dispositionality, while the latter implies transient behavior not necessarily informative of one's personality. This biased language use can maintain or even strengthen cognitive group biases in a self-perpetuating cycle. Given that language abstraction is associated with cognitive generalization of behaviors to the group level [35], maintaining or accentuating a positive depiction of the in-group and a negative depiction of the out-group is likely a consequence.

Mirroring the mechanisms of in-group favoritism and out-group derogation, preconceived political attitudes can alter the processing of political content, such as videos of speeches [8] and statements [36–38], in a predictable way. Taber and Lodge [37] speak of the presence of both a disconfirmation and a confirmation bias. The former refers to people readily accepting attitudinally congruent information, while skeptically counterarguing incongruent evidence. The latter describes the tendency to actively seek non-threatening, confirming information out of a palette of provided evidence rather than weighing pros and cons. Consonantly, based on the observation that preconceived political ideology negatively affected syllogistic reasoning for political syllogisms, Keller and colleagues [36] argue that political ideology impairs logical reasoning.

To summarize, the perception of speech can be influenced by a multitude of characteristics of both the speaker and the recipient, such as gender [3,4], political attitude [8] and more. In political discourse, a phenomenon of favoring male political speakers over female pendants is well established and described by RCT. Further, SIT posits that humans are generally prone to in-group favoritism and out-group derogation, which can even be observed on a lingual level. The current study therefore attempts to fill a gap in literature by investigating the intricate interactions imposed by both theories on political speech perception on a very basal linguistic level: lexemes. In this research study, we therefore assessed the interpretation of German lexemes with a political background, while taking political ideology as well as speaker and recipient gender into account. Conceptually, the set of lexemes was created to reflect the vast ranges of political speech: from neutral to highly pejorative, and from stereotypical left-wing to stereotypical right-wing vocabulary. The lexemes were evaluated in terms of their perceived political connotations on a left-right spectrum and pejorative weight, i.e., the

degree to which a particular word is considered offensive. We expect that the gender of both the speaker and the recipient influences the evaluation of the perceived pejorative weight of the words beyond the impact of political attitude. Since we could not find any comparable studies, i.e., empirical research exploring the processing of a diverse set of political language tokens with respect to group identity, the justifiability of formulating empirically driven hypotheses was questionable. Hence, we pursued a theoretical derivation of hypotheses by drawing from RCT as well as SIT: We assumed that the in-group favoritism (proposed by SIT) gets amplified by the male-favoritism (proposed by RCT) in male participants, while the in-group favoritism in female participants gets cancelled out by the male-favoring RCT effect. Thus, we hypothesized that men rate political lexemes produced by male producers as less derogatory and politically less extreme/more neutral than when produced by female pendants, while women do not display any kind of favoritism.

## Methods

### Participants and anonymity

Ninety-five participants participated in this online survey. Thereof, three reported that they identified neither as male nor female. Since these participants provided too few data for meaningful statistical analyses as an own group, they were excluded from the analyses. This led to a final sample of ninety-two participants between the ages of 18 and 64 ($M = 25.18$, $SD = 8.50$) with 46 male and 46 female participants. Power calculations were based on the effects reported by two studies reporting both participant gender differences and speaker gender differences in terms of the perceived offensiveness of slurs [3,4]. The two studies reported medium to large effect sizes for both contrasts. As a result, prospective sample size calculation considering a Cohen's $f^2$ of .30, a power of .80 and 4 groups yielded a minimum sample size of 90, as indicated by the G*Power software (version 3.1.9.7) [39]. In terms of highest education, 61 participants reported having a high-school diploma and 31 reported having a university degree. Concerning socio-political ideology, as quantified by the Social Dominance Orientation Questionnaire (SDO-7) [22,40], the sample showed a mean Social Dominance Orientation (SDO) value of 2.45 ($SD = 0.91$) in a possible range of 1 (low Social Dominance Orientation) to 7 (high Social Dominance Orientation). For more information on the SDO, see section 'Survey Procedure' below. Participants were recruited by means of a university internal email distribution list and university internal forums. German as first language and a minimum age of 18 were the inclusion criteria. The survey was conducted in a fully anonymous manner from February 14th to May 22nd 2024. Due to the online environment, all participants gave their informed consent by button click. All procedures of the present study have been conducted in accordance with the Declaration of Helsinki and were approved by the ethics committee of the University of Graz (GZ. 39/172/63 ex 2022/23). The procedures of this study were pre-registered at the Open Science Framework (OSF; https://osf.io/8c53w; deviations from the pre-registered methods are listed in S2 Table).

### Stimuli

**Lexeme pool.** We curated the initial lexeme pool by drawing 'seed words' from previous literature [41–43], which functioned as topic determinants. Scharloth [44] as well as Bernstorff [43] provided lexica of words linked to right-wing rhetorics. These represent comprehensive works, which we could draw vast number of seed words from. Resources of left-wing lexis on the other hand were scarce: Pertsch [41] provided a lexicon of lexemes associated with diversity, which is not to be equated with left-wing ideology per se, but provided a number of useful seed words anticipatedly used by left-wing individuals. Seed words were grouped by political topic. Since the seed words frequently were politically laden and pejorative, we attempted to fill the gaps in the two-dimensional continuum created by pejorative weight and political connotation by providing neutral pendants. The rationale behind this approach was to balance the distribution of semantic concepts across the ranges of these dimensions. Examples are provided in the following: 1. The topic of immigration: 'Invasion' (en: 'invasion') [42,43], 'Migration' (en: 'migration') [43], 'Ausländerflut' (en: 'flood of foreigners') [43] and 'Einwanderung' (en: 'immigration') [42]. 2. The topic of homosexuality: 'Schwuler' (en: 'gay person') [41,42], 'Schwuchtel' (en: 'faggot') [42], and 'Homosexuelle Person' (en: 'homosexual person'). The latter was added as an anticipatedly

non-derogatory pendant. We aimed at covering a multitude of topics such as immigration, social welfare, climate protection etc. Finally, as we observed an overrepresentation of right-wing pejoratives, we attempted to add pejoratives anticipatedly associated with the political mid or left, such as 'Waffenfanatiker' (en: 'gun maniac'), 'Hetzer' (en: 'political agitator') and 'Aluhutträger' (en: a person wearing a tin foil hat). Ultimately, we ended up with 122 lexemes with a political background (S1 Table) covering a broad range of political topics.

### Avatars

To introduce the gender of the fictive producers of the lexemes, avatars were used, which were accompanied by the following description: "This is [Sam/Alex]. [He/She] votes exclusively for [left/right] parties. Concerning all relevant political topics, [he/she] shares [leftist, liberal/rightist, conservative] points of view and represents those in discussions.". See Fig 1A for the original German description. The avatars were rudimentary, pictographic versions of gendered silhouettes and were positioned at the poles of the political connotation dimension (Fig 1B). Participants either exclusively saw female or male avatars. Avatar gender was balanced within each participant gender group.

### Pejorative weight and political connotation along a left-right spectrum

The perceived political connotation on a left-right spectrum was evaluated by answering the question "Who would rather use this word?" on a visual analogue scale (VAS) ranging from 0 (left wing avatar) to 100 (right wing avatar). Although the common left-right dimension repeatedly faced criticism arguing that it represents an oversimplification of a multidimensional construct and shows profound regional variations of the meanings of 'left' and 'right' [45,46], it represents a popular, well established measure that is widely used to date [47–50]. Despite the notion that real-world political attitudes, policies, and issues appear to be substantially more complex, a unidimensional left–right measure is considered particularly useful for capturing individuals' subjective socio-cognitive perceptions of political phenomena [51], and is considered successfully discriminative in terms of the social dominance orientation [40,52,53]. Evidence from earlier research on the perceived allocation of political parties along a left-right spectrum attributed superior reliability to ordinal Likert-compared to VAS, potentially due to a reduction of cognitive burden for the raters [48]. Despite that, we opted for a VAS format, as we wanted to ensure the detection of fine-grained differences between lexemes. The pejorative weight of the lexemes was indicated by answering the question "How derogatory do you find the word?" on a VAS ranging from 0 ("not derogatory") to 100 ("very derogatory"). See Fig 1A for an original German example. The 'pejorative weight' of lexemes, as we call it throughout this manuscript, appears in the literature under a variety of names, such as 'force' [54] or 'level of emotion' [55], all of which measure the degree to which lexemes are considered derogatory/insulting.

### Social dominance orientation (SDO)

In the current study we assessed participants' Social Dominance Orientation (SDO) by administration of the SDO-7 questionnaire [40,52]. Social dominance orientation is a personality trait that reflects the tendency to think of one's own group as superior compared to others. It correlates significantly with several ideologies, such as nationalism, anti-black racism and elitism, as well as attitudes towards homosexual rights, social programs and environmental policies [52]. We therefore used the SDO as a measure of political attitude in the current study.

### Survey procedure

First, participants gave their informed consent. Then, participants answered socio-demographic questions comprising age, gender, first language, education level and occupation. Participants then indicated which words of a list of 122 German lexemes with a political background (S1 Table) they were familiar with. Subsequently, they evaluated these lexemes in

# A

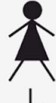

Zu diesem Zweck repräsentieren folgende Figuren die politischen Extrema:

Das ist Alex, sie wählt ausschließlich linke Parteien. Sie teilt hinsichtlich aller relevanten politischen Themen linke, liberale Standpunkte und vertritt diese in Gesprächen.

Das ist Sam, sie wählt ausschließlich rechte Parteien. Sie teilt hinsichtlich aller relevanten politischen Themen rechte, konservative Standpunkte und vertritt diese in Gesprächen.

# B

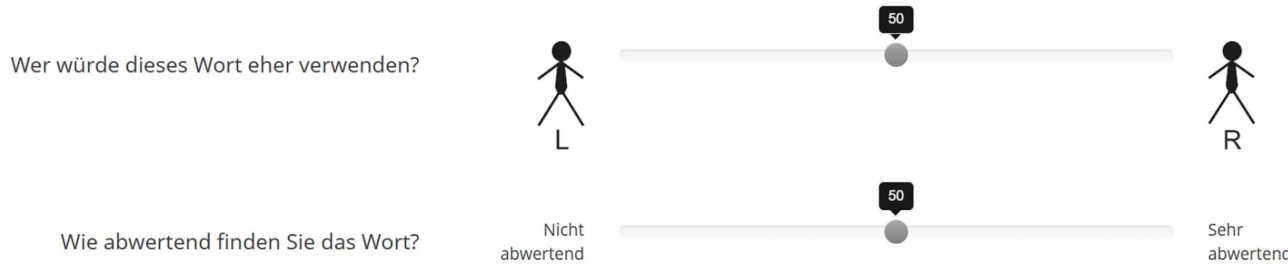

Wer würde dieses Wort eher verwenden?

Wie abwertend finden Sie das Wort?    Nicht abwertend    Sehr abwertend

**Fig 1. Avatar descriptions and an example item. A.** German avatar descriptions based on female avatars. Translation: I: "For this cause, the following avatars represent the political extremes: This is Alex, she exclusively votes for left-wing parties. Concerning all relevant political topics, she has left-wing, liberal points of view and represents them in discussions.". II: "This is Sam, she exclusively votes for right-wing parties. Concerning all relevant political topics, she has right-wing, conservative points of view and represents them in discussions.". **B.** Rating scale for political connotations based on male avatars and pejorative weight. Translation: I: "Who would rather use this word?". II: "How derogatory do you find the word?". VAS from 0 = "Not derogatory" to 100 = "Very derogatory".

terms of the perceived political connotation and the pejorative degree (Fig 1B). Finally, the Social Dominance Orientation Questionnaire (SDO7) [40] was administered.

## Preprocessing and statistical analyses

All preprocessing and statistical analyses were performed within R (version 4.4.2). We employed k-means clustering to find clusters of dense data. We derived the optimal k of 4 by means of an elbow plot (Fig 2A). The four extracted clusters roughly reflected all possible combinations of poles across both variables, i.e., politically left & less derogatory (L-low), politically right & less derogatory (R-low), politically left & derogatory (L-high), politically right & derogatory (R-high; Fig 2B). To

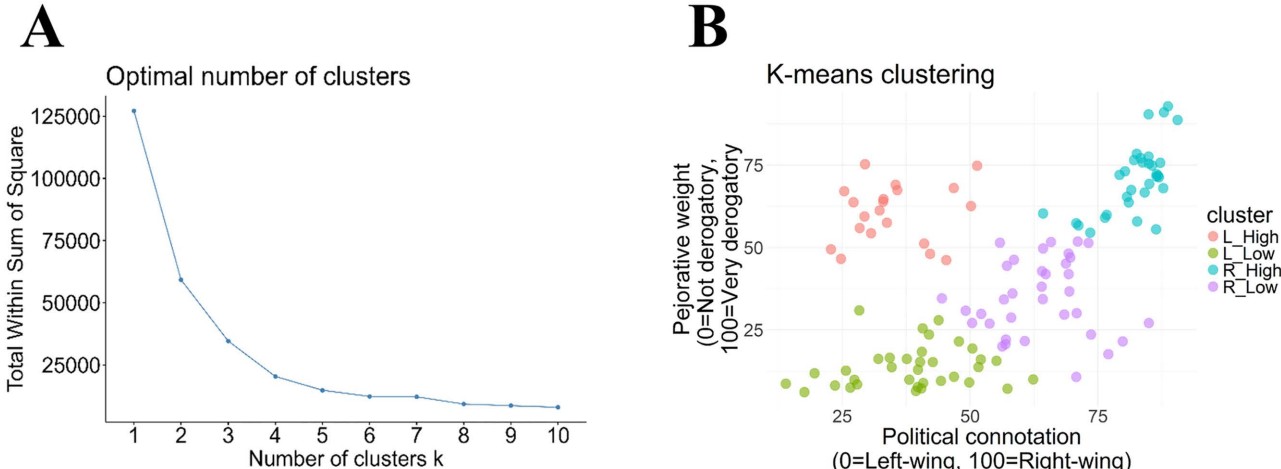

**Fig 2. Justification and visual display of lexeme allocation when using k = 4 classes. A.** Elbow plot showing total within sum of squares per number of clusters **k. B.** All data points (lexemes) on a 2d graph (political connotation vs. pejorative weight), color-coded by cluster.

determine whether *Age* and mean *SDO* values differ across groups, between-subjects analysis of variance (ANOVA) was conducted for both variables separately.

For the main analysis we performed generalized linear mixed effects models on each of the clusters by using the "glmmTMB" package (version 1.1.11) [56] in R. Models were constructed for all word clusters separately. While *pejorative weight* represented the response variable (RV) in one domain of analyses, *political connotation* was used in the second one, yielding 8 (4 clusters * 2 domains) models in total. *Participant gender* and *avatar gender* functioned as categorical predictors. The random effects structure of all models consisted of the mean *SDO* scores, encoding participants, and *words,* encoding the used lexemes. As a starting point, the lognormal distribution with the corresponding logarithmic link function was set as the underlying distribution family for each model. The two "Low" (R-Low, L-Low) clusters of the political connotation branch of analyses required a regular Gaussian distribution as indicated by the histograms of the RVs as well as diagnostic plots produced by the "DHARMa" package (version 0.4.7). Therefore, these models were equivalent to regular linear mixed effects models. Histograms of our lognormal models indicated zero-inflation which was accounted for by setting the "ziformula" argument of the glmmTMB function to "~1". The Gaussian models did not require zero-inflation correction. QQ-plots were used to check the distributional assumptions of the model regarding both the residuals and the random effects. In terms of expected residual distribution, the R-Low cluster in the pejorative weight domain as well as the R-Low and the L-Low cluster in the political connotation domain showed slight deviations. In terms of normality of random effects, the R-High cluster in the pejorative weight domain as well as the R-Low and the L-Low cluster in the political connotations domain showed slight deviations. To assess main effect and interaction significance, Type III Wald Chi-Square tests were performed. Post-hoc comparisons were performed by means of the "emmeans" package (version 1.11.0). Contrary to conventional ANOVAs and related statistical models, post-hoc contrasts for linear mixed models perform on model-predicted data, meaning that post-hoc contrasts are computed based on model output and not raw data. Alpha error correction regarding post-hoc contrasts was performed by applying the Bonferroni-Holm method. In line with [57], we performed post-hoc tests irrespective of main effect and interaction significance. This specifically affects the R-Low model in the pejorative weight domain, as it produced neither significant main effects nor a significant interaction (S3 Table). All preprocessing steps, model assumptions, analyses can be further examined by accessing the provided online materials (https://osf.io/sgbqf) encompassing all participant data and the processing script.

## Results

In the following, the results pertaining to the pejorative RV will be interpreted differently from the political RV. In opposition to pejorative weight, which can be interpreted irrespective of value range, since a higher value on the scale always corresponds to a higher pejorative degree, the interpretation of group differences concerning political connotation greatly relies on the respective value range. Therefore, we interpret these results in terms of extremity, with group means approaching the poles of the scale (0 - "left-wing", 100 - "right-wing") considered as more extreme/having a stronger political connotation and values approximating the middle value of the scale, 50 (neither left- nor right-wing), as less extreme/having a weaker political connotation.

### Sample characteristics

In terms of Social Dominance Orientation (SDO) the four groups, female recipients with female producers of political lexemes (FF), female recipients with male producers (FM), male recipients with female producers (MF), and male recipients with male producers (MM), showed no significant differences ($p > .05$).

### Main analysis

The linear mixed effects model results in the pejorative weight domain are portrayed in Fig 3 and Table 1. The linear mixed effects model results in the political connotation domain are portrayed in Fig 4 and Table 2. In addition to z- and p-values, estimated marginal means and corresponding standard errors are reported. The significance threshold was set at 0.05.

### Pejorative weight: R-Low model

FF participants ($\hat{x} = 53.37$, $SE = 3.43$) rated R-Low words significantly more derogatory as compared to the FM group ($\hat{x} = 45.53$, $SE = 2.95$, $z_{inf} = -3,98$, $p < .001$), the MF group ($\hat{x} = 46.05$, $SE = 3.07$, $z_{inf} = -2.79$, $p = .021$) and the MM group ($\hat{x} = 43.03$, $SE = 2.76$, $z_{inf} = -5.25$, $p < .001$). All remaining contrasts were non-significant at significance level $\alpha = .05$.

### Pejorative weight: L-Low model

Both male and female recipients rated L-Low words as less derogatory when the producer was male (MM: $\hat{x} = 24.50$, $SE = 1.79$; FM: $\hat{x} = 23.49$, $SE = 1.75$) as compared to when the producer was female (MF: $\hat{x} = 31.71$, $SE = 2.39$; FF: $\hat{x} = 27.88$, $SE = 1.99$; MM-MF: $z_{inf} = -3.51$, $p = .002$; FM-FF: $z_{inf} = -3.27$, $p = .004$). Further, the FM group perceived L-Low words as less derogatory than the MF group ($z_{inf} = -3.90$, $p < .001$). All remaining contrasts were non-significant at significance level $\alpha = .05$.

### Pejorative weight: R-High model

The MM group ($\hat{x} = 55.31$, $SE = 3.06$) estimated the pejorative weight of R-High words significantly lower than all other groups: FM ($\hat{x} = 71.44$, $SE = 2.11$, $z_{inf} = 9.495$, $p < .001$), FF ($\hat{x} = 68.25$, $SE = 2.29$, $z_{inf} = 7.68$, $p < .001$), and MF ($\hat{x} = 61.31$, $SE = 2.77$, $z_{inf} = 3.19$, $p = .003$). Further, the FF group displayed significantly lower ratings than the FM group ($z_{inf} = -2.16$, $p = .031$) and higher ratings than the MF group ($z_{inf} = 3.84$, $p < .001$). Finally, the MF group produced lower ratings than the FM group ($z_{inf} = -5.41$, $p < .001$).

### Pejorative weight: L-High model

In terms of L-High words, the MM group ($\hat{x} = 46.07$, $SE = 3.29$) reported a lower pejorative weight compared to all other groups: MF ($\hat{x} = 56.34$, $SE = 2.86$, $z_{inf} = 4.33$, $p < .001$), FM ($\hat{x} = 64.67$, $SE = 2.37$, $z_{inf} = 9.06$, $p < .001$) and FF ($\hat{x} = 56.20$, $SE = 2.89$, $z_{inf} = 4.54$, $p < .001$). FM individuals reported a higher pejorative weight compared to both MF ($z_{inf} = -3.60$, $p < .001$) and FF ($z_{inf} = -4.23$, $p < .001$) counterparts. The remaining contrast was non-significant at significance level $\alpha = .05$.

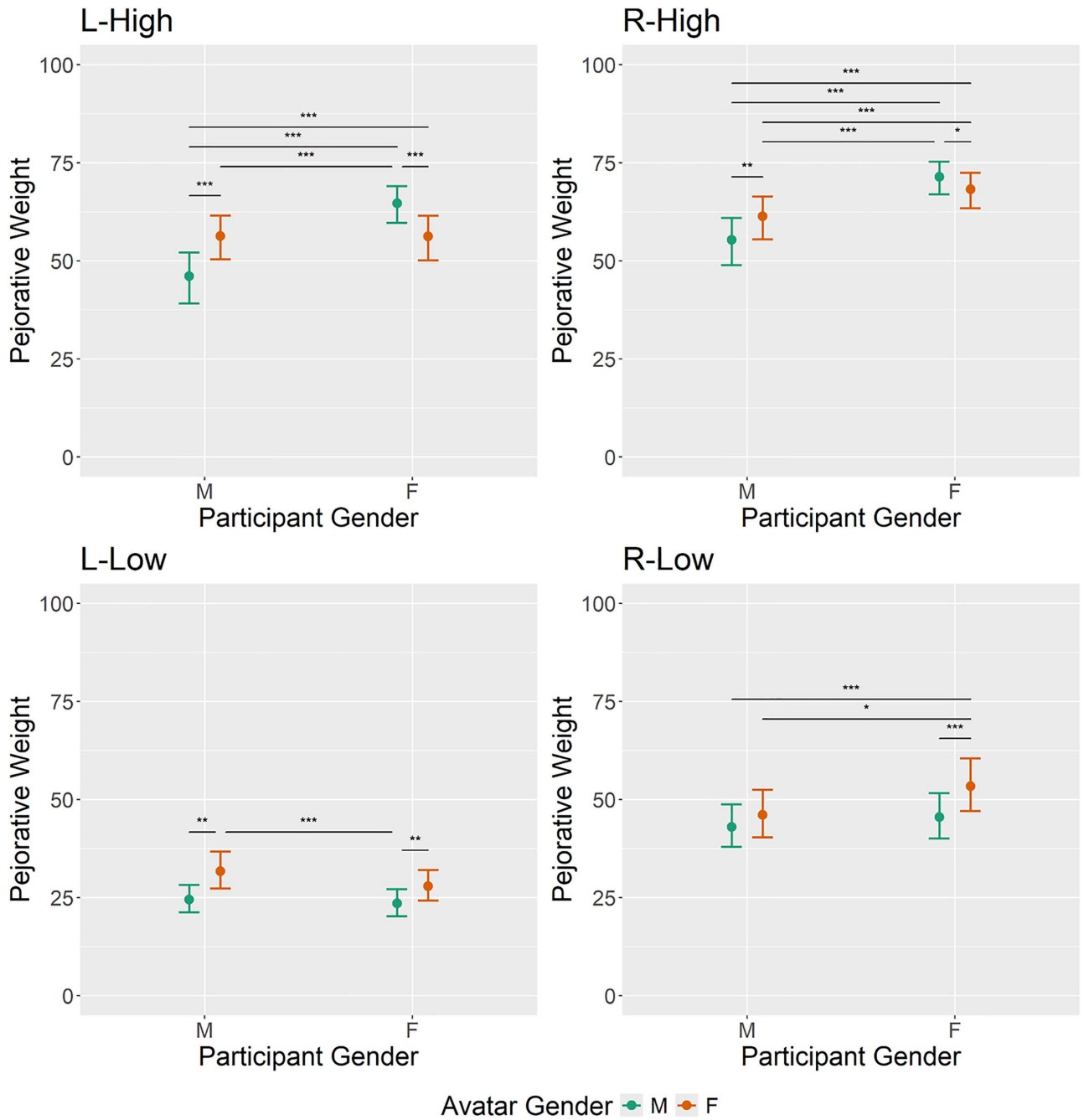

**Fig 3. Linear mixed effects model results in the pejorative weight domain for each lexeme cluster.** Error bars indicate 95% confidence intervals of the estimated marginal means.

### Political connotation: R-Low model

R-low lexemes were rated as less extreme by MM individuals ($\hat{x} = 60.6$, $SE = 1.94$) compared to MF ($\hat{x} = 64.89$, $SE = 1.98$, $t_{2981} = -2.86$, $p = .017$) and FF ($\hat{x} = 68.00$, $SE = 1.97$, $t_{2981} = -5.30$, $p < .001$) pendants, respectively. Furthermore, the FM

**Table 1. Linear mixed effects model results pertaining to the z-standardized response variable pejorative weight on the basis of each lexeme cluster and all lexemes combined.**

| Cluster | Contrast | Estimate | SE | df | z-ratio | p |
|---|---|---|---|---|---|---|
| R-Low | M M – F M | −0.0563 | 0.0419 | Inf | −1.344 | .537 |
| | M M – M F | −0.0677 | 0.0508 | Inf | −1.332 | .537 |
| | M M – F F*** | −0.2153 | 0.041 | Inf | −5.249 | <.001 |
| | F M – M F | −0.0114 | 0.0544 | Inf | −0.21 | .834 |
| | F M – F F*** | −0.159 | 0.04 | Inf | −3.978 | <.001 |
| | M F – F F* | −0.1476 | 0.053 | Inf | −2.787 | .021 |
| L-Low | M M – F M | 0.042 | 0.0576 | Inf | 0.730 | 0.465 |
| | M M – M F** | −0.2578 | 0.0734 | Inf | −3.514 | 0.002 |
| | M M – F F | −0.1293 | 0.0544 | Inf | −2.376 | 0.053 |
| | F M – M F*** | −0.2998 | 0.0769 | Inf | −3.900 | <.001 |
| | F M – F F** | −0.1714 | 0.0524 | Inf | −3.273 | 0.004 |
| | M F – F F | 0.1284 | 0.0708 | Inf | 1.814 | 0.139 |
| R-High | M M – F M*** | 0.4475 | 0.0471 | Inf | 9.495 | <.001 |
| | M M – M F** | 0.1441 | 0.0451 | Inf | 3.193 | 0.003 |
| | M M – F F*** | 0.3419 | 0.0445 | Inf | 7.678 | <.001 |
| | F M – M F*** | −0.3033 | 0.0561 | Inf | −5.410 | <.001 |
| | F M – F F* | −0.1056 | 0.0488 | Inf | −2.163 | 0.031 |
| | M F – F F*** | 0.1978 | 0.0515 | Inf | 3.839 | <.001 |
| L-High | M M – F M*** | 0.423 | 0.0467 | Inf | 9.061 | <.001 |
| | M M – M F*** | 0.2114 | 0.0489 | Inf | 4.325 | <.001 |
| | M M – F F*** | 0.208 | 0.0458 | Inf | 4.540 | <.001 |
| | F M – M F*** | −0.2116 | 0.0588 | Inf | −3.600 | <.001 |
| | F M – F F*** | −0.215 | 0.0509 | Inf | −4.225 | <.001 |
| | M F – F F | −0.0034 | 0.0561 | Inf | −0.061 | 0.9518 |

*p < .05, ** p < .01, *** p < .001

group ($\hat{x} = 61.57$, $SE = 1.98$) produced less extreme ratings than the FF group ($t_{2981} = −4.62$, $p < .001$). All remaining contrasts were non-significant at significance level $\alpha = .05$.

### Political connotation: L-Low model

The MM group ($\hat{x} = 41.27$, $SE = 2.17$) considered L-Low words politically significantly less extreme compared to the FF ($\hat{x} = 36.21$, $SE = 2.19$, $t_{3138} = 3.177$, $p = .008$) and FM group ($\hat{x} = 37.23$, $SE = 2.19$, $t_{3138} = 4.00$, $p < .001$). All remaining contrasts were non-significant at significance level $\alpha = .05$.

### Political connotation: R-High model

The MM group ($\hat{x} = 72.26$, $SE = 1.61$) considered the political connotation of R-High words significantly less extreme than all other groups: FM ($\hat{x} = 79.36$, $SE = 1.27$, $z_{inf} = 6.88$, $p < .001$), FF ($\hat{x} = 80.27$, $SE = 1.21$, $z_{inf} = 8.04$, $p < .001$), and MF ($\hat{x} = 79.59$, $SE = 1.28$, $z_{inf} = 6.30$, $p = .003$). All remaining contrasts were non-significant at significance level $\alpha = .05$.

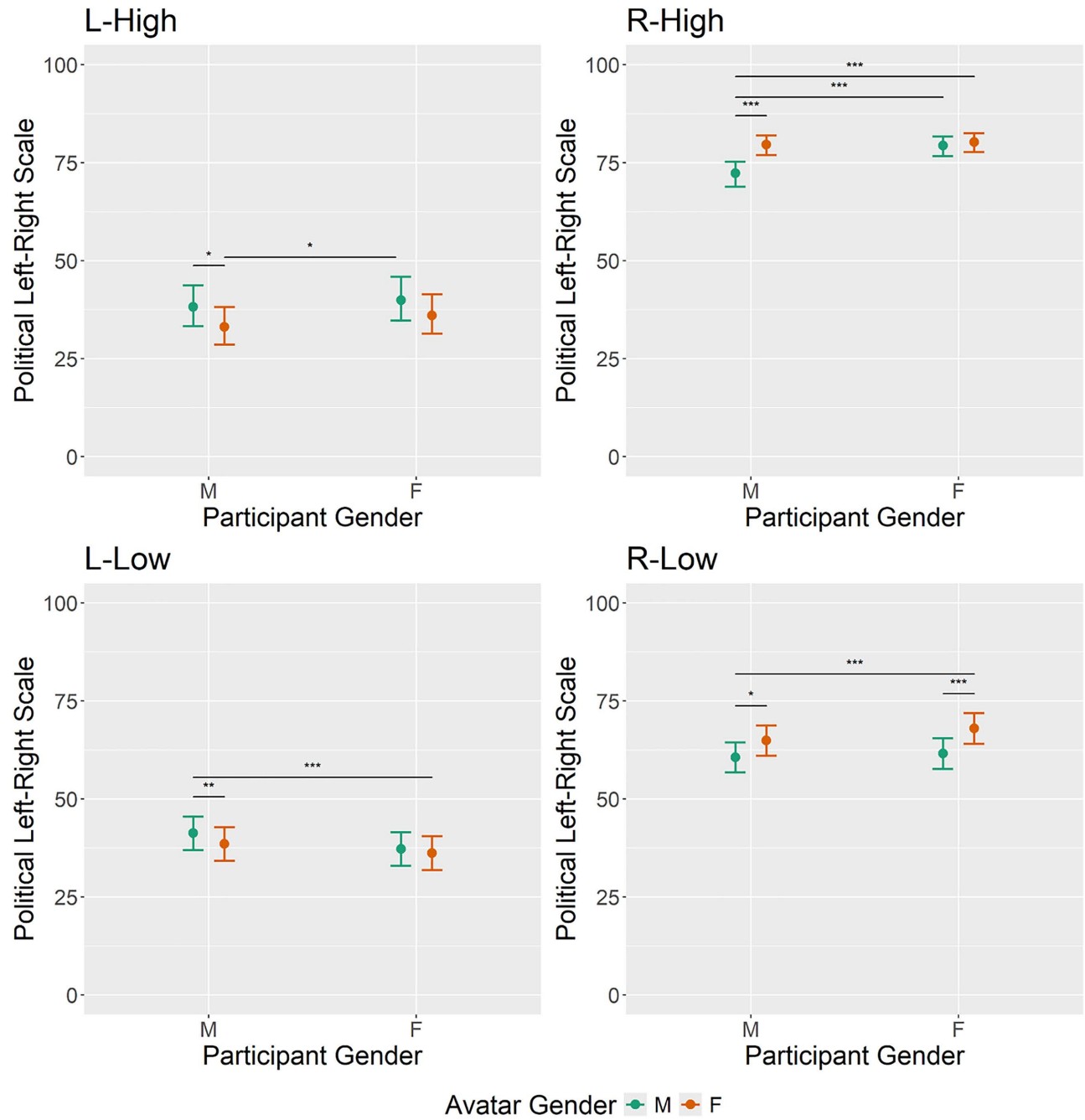

**Fig 4. Linear mixed effects model results in the political connotation domain for each lexeme cluster.** Error bars indicate 95% confidence intervals of the estimated marginal means.

### Political connotation: L-High model

Both groups which were confronted with a male producer – FM ($\hat{x}=39.92$, $SE=2.85$) and MM ($\hat{x}=38.21$, $SE=2.64$) – rated L-High words as less extreme than the MF group ($\hat{x}=33.10$, $SE=2.44$, FM-MF: $z_{inf}=3.05$, $p=.014$; MM-MF: $z_{inf}=2.59$, $p=.048$). All remaining contrasts were non-significant at significance level $\alpha=.05$.

**Table 2. Linear mixed effects model results pertaining to the z-standardized response variable political connotation on the basis of each lexeme cluster and all lexemes combined.**

| Cluster | Contrast | Estimate | SE | df | z-ratio | p |
|---|---|---|---|---|---|---|
| R-Low | M M – F M | −0.9356 | 1.4162 | 2981 | −0.6607 | .509 |
| | M M – M F* | −4.2523 | 1.4846 | 2981 | −2.8644 | .017 |
| | M M – F F*** | −7.37 | 1.3906 | 2981 | −5.2999 | <.001 |
| | F M – M F | −3.3167 | 1.6355 | 2981 | −2.028 | .128 |
| | F M – F F*** | −6.4344 | 1.3918 | 2981 | −4.6231 | <.001 |
| | M F – F F | −3.1177 | 1.5541 | 2981 | −2.006 | .128 |
| L-Low | M M – F M** | 4.0363 | 1.2706 | 3138 | 3.1766 | .008 |
| | M M – M F | 2.7197 | 1.3262 | 3138 | 2.0508 | .162 |
| | M M – F F*** | 5.0636 | 1.2666 | 3138 | 3.9976 | <.001 |
| | F M – M F | −1.3167 | 1.4365 | 3138 | −0.9166 | .719 |
| | F M – F F | 1.0272 | 1.2466 | 3138 | 0.824 | .719 |
| | M F – F F | 2.3439 | 1.3852 | 3138 | 1.6921 | .272 |
| R-High | M M – F M*** | 0.2953 | 0.0429 | Inf | 6.8813 | <.001 |
| | M M – M F*** | 0.3068 | 0.0487 | Inf | 6.302 | <.001 |
| | M M – F F*** | 0.3408 | 0.0424 | Inf | 8.0429 | <.001 |
| | F M – M F | 0.0115 | 0.0534 | Inf | 0.2156 | .999 |
| | F M – F F | 0.0455 | 0.0443 | Inf | 1.0269 | .913 |
| | M F – F F | 0.034 | 0.0505 | Inf | 0.6731 | .999 |
| L-High | M M – F M | −0.0439 | 0.048 | Inf | −0.9136 | .451 |
| | M M – M F* | 0.1434 | 0.0553 | Inf | 2.5905 | .048 |
| | M M – F F | 0.0578 | 0.0457 | Inf | 1.2657 | .451 |
| | F M – M F* | 0.1872 | 0.0615 | Inf | 3.0464 | .014 |
| | F M – F F | 0.1017 | 0.0475 | Inf | 2.1425 | .129 |
| | M F – F F | −0.0856 | 0.0595 | Inf | −1.4387 | .451 |

*p < .05, ** p < .01, *** p < .001

## Discussion

In the current study we explored whether the gender of the producer of political lexemes as well as the gender of the recipient influences how lexemes with a political background are interpreted. We anticipated that male participants would rate political lexemes produced by men as less derogatory and less politically extreme compared to women producing said lexemes. For female participants, we expected no producer-dependent differences. For this reason, political expressions were evaluated by means of two dimensions: subjective political connotations on a left-right spectrum and pejorative weight. Statistical analyses were performed on four different word clusters, characterized by different expressions in pejorative weight and political connotation: right-wing words with low pejorative weight (R-Low), left-wing words with low pejorative weight (L-Low), right-wing words with high pejorative weight (R-High), and left-wing words with high pejorative weight (L-High). In three of the four word clusters (R-High, L-High, L-Low) male participants rated political lexemes as less derogatory when the producer was male. In the remaining word cluster (R-Low) the same trend was visible, yet failed to reach significance. Therefore, male recipients were less sensitive/more forgiving when the producer of a word belonged to the same gender as indicated by the lower ratings in pejorative weight. Female participants produced a more complex pattern: When confronted with words belonging to the R-Low or L-Low cluster, female participants showed higher tolerance towards male producers. Words belonging to the R-High or L-High cluster, on the other hand, led female recipients

to showing a higher tolerance towards the in-group, as indicated by lower pejorative ratings when the producer of a word was female. In terms of political connotation, male participants showed the same elevated tolerance towards in-group producers of words: across three of the four word clusters (R-High, L-High, R-Low), words produced by a male avatar were rated as politically less extreme (closer to the center of the scale) compared to female avatars. In the remaining word cluster (L-Low) the same trend was visible, yet failed to reach significance. Consonantly, an increased tolerance towards male producers of politically charged words also emerged in female recipients, but only in the R-Low cluster. The remaining word clusters did not yield significant results. In the following we elaborate on our results by providing a theory-driven explanation.

Drawing from Role Congruity Theory (RCT) and Social Identity Theory (SIT), our hypothesis was that men would exhibit in-group favoritism, while women would not display any favoritism. We expected this to manifest in men generally rating lexemes produced by men as less derogatory and more neutral/less extreme in terms of political connotation compared to those produced by female speakers. The analysis of political connotations generally supported our expectations: As predicted, the absence of producer gender effects in the female participant groups regarding political connotations on a left-right spectrum can be explained by a mutual cancellation of male-favoritism, as proposed by the RCT, and the in-group favoritism, as proposed by the SIT. Furthermore, in line with our hypotheses male recipients favored in-group producers of words, as evidenced by politically less extreme/more neutral ratings. This may be explained by a mutual reinforcement of the male-favoritism (RCT) and the in-group favoritism (SIT). Regarding pejorative weight, a more complex pattern emerged, particularly in female participant groups. In accordance with our considerations, male recipients displayed comparable in-group favoritism, as evidenced by words produced by male avatars being rated less derogatory than those produced by female avatars in three out of four word clusters (R-High, L-High, L-Low). In contrast, female recipients favored male producers in less derogatory clusters (R-Low, L-Low), while favoring female producers in highly derogatory clusters (R-High, L-High). It appears that male favoritism generally dominates in-group favoritism in terms of the interpretation of less derogatory words, whereas the opposite holds true for highly derogatory word clusters in the pejorative weight domain.

RCT suggests that people generally favor leaders whose gender identities align with their demonstrated traits [14,15]. The inability of women to satisfy the prescribed normative gender expectations, such as warmth and sensitivity, while simultaneously showing characteristic leadership traits, such as dominance and assertiveness, puts them in a double bind. This ultimately causes female leaders and politicians to be perceived as less competent [15], less likeable [15] and more aggressive [7]. Overall, the results of the current study are indicative of male favoritism with regards to less derogatory political language, i.e., L-Low and R-Low lexeme clusters, irrespective of recipient gender. Drawing from RCT, this finding suggests that both men and women are prone to favoring male producers of less derogatory, written political lexemes due to gender role congruencies. In more derogatory word clusters, i.e., L-High and R-High, the emerging response pattern is more complex, as male recipients generally favored male producers, while female recipients favored female producers only in the pejorative weight domain and showed no favoritism with respect to political connotations. While there is no evidence against the presence of an identical role-congruency-based male favoring mechanism in more derogatory lexeme clusters, our results indicate that in-group favoritism is either dominating or exclusively driving the responses to derogatory political language in the pejorative weight domain. This is attributed to the SIT which we elaborate on in the following paragraph.

According to SIT, intergroup bias is driven by the need to create, restore or uphold a positive social identity. Intergroup bias is even more pronounced when the social identity, e.g., the group status, is threatened [30]. The processes behind linguistic intergroup bias (LIB) [32] perfectly illustrate this: abstract descriptions are activated when in-group members show desirable behavior or out-group members show non-desirable behavior (non-threat condition), implying temporal stability of the actions. Conversely, concrete descriptions are activated when in-group members show non-desirable

behaviors and out-group members show desirable behaviors (threat condition), indicating temporal instability. Cognitive inter-group bias was experimentally shown to get more pronounced when introducing threat to the group integrity/social identity [31,32,58]. Analogously to the linguistic inter-group bias, such a group-serving attribution bias might partly explain the results of the current study: When participants rated derogatory language produced by in-group members, i.e., individuals of the same gender in this case, they may have been susceptible to cognitive inter-group bias: Analogously to attributing transience to non-desirable in-group behavior, as displayed by the LIB mechanism, individuals may have been persuaded to tolerate derogatory language from in-group members to a larger degree in order to uphold a positive social identity [30]. Simultaneously, individuals may be less tolerant of derogatory language produced by out-group people due to a lower level of identification and no need for social identity protection, resulting in higher ratings. This may explain why only highly derogatory lexemes activated in-group favoring behavior in the pejorative weight domain. However, for the sake of completeness, despite the evidence supporting the notion that default male favoritism gets overridden by in-group favoritism in both gender groups, we want to emphasize that there exists an alternative explanation: Since the results do not allow for discrimination between default male favoritism and in-group favoritism in male participants as they are latent processes which identically manifest in the data, it is also valid to conclude that men might show no protective mechanism to support a positive social identity at all, but rather show a consistent default behavior emerging as male-favoritism, driven by role congruity. In this case, protective in-group favoritism would be specific to women. However, since there is no argument to believe that only women are prone to protective in-group favoring behavior upon confrontation with derogatory political language, we consider our original interpretation more plausible.

In summary, male favoritism, as predicted by the RCT, appears to represent a default response to less derogatory political language, irrespective of the recipient's gender. However, when political language becomes more derogatory, in-group favoritism (SIT) emerges as a protective mechanism for maintaining social identity, effectively overriding male favoritism. Notably, this pattern was observed exclusively within the domain of pejorative weight. Regarding political connotations along the left-right spectrum, female recipients exhibited no in-group favoritism in response to more derogatory lexeme clusters, while male recipients showed very consistent male-favoritism across all clusters.

### Limitations and future directions

In the current study, the investigated set of stimuli consisted of isolated written lexemes produced by fictional speakers. In order to draw broadly generalizable, collective conclusions about the proposed effects, future studies are advised to investigate the mechanism explored in the current study in the context of spoken language. Further, future research is advised to investigate gender effects in response to political language in a larger discourse context by weaving critical words into statements or entire speeches. Furthermore, we observed non-significant trends suggesting male favoritism among female recipients in the domain of political connotations. These trends closely resemble the significant response patterns observed in their male counterparts. This suggests that the effects may be smaller at the population level, potentially necessitating a larger sample size to achieve sufficient statistical power for detection. Finally, future research might aim at disentangling the effects of SIT and RCT. This could potentially be achieved by comparing non-gendered social groups in terms of their perception of political language produced by in-group vs. out-group members. To further substantiate the existence of protective in-group favoritism irrespective of gendered effects, we would anticipate lower pejorative weight ratings for in-group producers than for out-group producers when in response to derogatory political language, while non-derogatory political language should yield no differences.

### Conclusion

The present study primarily examined gender differences in the interpretation of derogatory and non-derogatory political language. Participants assessed politically charged lexemes based on two dimensions: their political connotations along a left-right spectrum and their pejorative weight. The analysis involved four groups, categorized by the gender of both the recipients and

producers of the political language: male recipients with male producers (MM), male recipients with female producers (MF), female recipients with male producers (FM), and female recipients with female producers (FF). The findings revealed that derogatory language elicited in-group favoritism, which manifested itself as numerically lower pejorative weight ratings when the producer was of the same gender. This was interpreted as raters being more forgiving when producers of political lexemes were in-group members. This is considered a protective mechanism aimed at safeguarding in-group identity – a phenomenon popularized by SIT. In contrast, a consistent pattern of male favoritism was observed irrespective of participant gender in less derogatory political language. However, when rating the political connotation of the lexemes along a left-right spectrum, women showed no such in-group favoring tendency, while men showed consistent male-favoritism across all word clusters.

## Supporting information

**S1 Table. Word list with corresponding clusters and mean values in pejorative weight and political connotation.**
(DOCX)

**S2 Table. Deviations from the pre-registration.**
(DOCX)

**S3 Table. Type III Wald Chi-Square test results displaying main and interaction effects of corresponding generalized linear mixed effects models.** PG = Participant gender, AG: Avatar gender. *p < .05, ** p < .01, *** p < .001.
(DOCX)

## Acknowledgments

The authors acknowledge the financial support by the University of Graz. The authors further acknowledge the support by the Field of Excellence COLIBRI (Complexity of Life in Basic Research and Innovation, University of Graz, Austria).

## Author contributions

**Conceptualization:** Edgar Onea, Silvia Erika Kober.

**Data curation:** Manuel Hons.

**Formal analysis:** Manuel Hons, Edgar Onea.

**Investigation:** Manuel Hons.

**Methodology:** Manuel Hons, Edgar Onea, Silvia Erika Kober.

**Project administration:** Manuel Hons.

**Resources:** Manuel Hons.

**Software:** Manuel Hons.

**Supervision:** Edgar Onea, Silvia Erika Kober.

**Validation:** Edgar Onea, Silvia Erika Kober.

**Visualization:** Manuel Hons.

**Writing – original draft:** Manuel Hons.

**Writing – review & editing:** Manuel Hons, Edgar Onea, Silvia Erika Kober.

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
