## [Decision Letter · Decision Letter 0]

23 Oct 2025

Dear Dr. Hons,

We look forward to receiving your revised manuscript.

Kind regards,

Xiaoming Tian, Ph.D.

Academic Editor

PLOS ONE

Journal Requirements:

2. Please note that your Data Availability Statement is currently missing [the repository name and/or the DOI/accession number of each dataset OR a direct link to access each database]. If your manuscript is accepted for publication, you will be asked to provide these details on a very short timeline. We therefore suggest that you provide this information now, though we will not hold up the peer review process if you are unable.

3. Please update your submission to use the PLOS LaTeX template. The template and more information on our requirements for LaTeX submissions can be found at http://journals.plos.org/plosone/s/latex .

“The authors acknowledge the financial support by the University of Graz. The authors further acknowledge the support by the Field of Excellence COLIBRI (Complexity of Life in Basic Research and Innovation, University of Graz, Austria)”

Reviewers' comments:

Reviewer's Responses to Questions

**Comments to the Author**

1. Is the manuscript technically sound, and do the data support the conclusions?

Reviewer #1: Yes

Reviewer #2: Partly

Reviewer #3: Partly

2. Has the statistical analysis been performed appropriately and rigorously?

Reviewer #1: Yes

Reviewer #2: Yes

Reviewer #3: I Don't Know

3. Have the authors made all data underlying the findings in their manuscript fully available?

Reviewer #1: Yes

Reviewer #2: Yes

Reviewer #3: Yes

4. Is the manuscript presented in an intelligible fashion and written in standard English?

Reviewer #1: Yes

Reviewer #2: No

Reviewer #3: Yes

Reviewer #1: I found the study timely, thought-provoking, and methodologically rigorous. I appreciate the effort to integrate Role Congruity Theory (RCT) and Social Identity Theory (SIT) in analyzing political language through a gendered lens.In the spirit of constructive academic dialogue, I would like to offer the following suggestions to enhance the clarity, coherence, and impact of your manuscript:

1.Language and Readability

Some phrases could be simplified for clarity and accessibility. For example, expressions such as

“evaluated German lexemes based on their subjective pejorative weight and political connotations along a left-right ideological spectrum” are precise but cognitively dense. Rephrasing for smoother readability may help reach a wider audience.

Additionally, certain sentences—such as the definition of Social Dominance Orientation (SDO) on line 189—are overly long. Consider breaking them into shorter units to aid reader comprehension.

2.Framing the Research Gap

In the introduction, it would strengthen the manuscript to more explicitly articulate the knowledge gap your study addresses. Clarifying how your research advances or diverges from existing work on linguistic intergroup bias and gendered political communication will underscore its theoretical significance.

3.Analytical Rationale

Beginning around line 136, where political connotation and pejorative weight are introduced as separate evaluative dimensions, it may be helpful to explain more clearly why these were analyzed independently. Elaborating on the conceptual or theoretical rationale behind this analytical separation would enhance the reader’s understanding of your design choices.

4.Methodological Clarifications

In the Methods section, please consider adding information on how participants were recruited and whether any eligibility criteria were applied. This would provide greater transparency and support replicability.

While the paper describes efforts to balance lexemes across ideological and pejorative categories, additional detail on how these items were pre-rated, selected, or validated would be beneficial.

5.Structure and Flow of the Discussion

The Discussion section would benefit from a clearer conceptual progression. A possible structure could be:

(1)A concise restatement of your hypotheses and summary of the key findings related to RCT.

(2)Interpretation of those findings specifically in terms of RCT’s implications.

(3)A subsequent shift to SIT, exploring how the results support, qualify, or complicate predictions about in-group favoritism and identity protection.

(4) A closing reflection on the broader implications of your findings—particularly how the interplay between RCT and SIT can inform future work on gender dynamics in political discourse.

Once again, I commend your interdisciplinary approach and the theoretical nuance embedded in your study. I hope these suggestions prove helpful as you revise and refine the manuscript.

Reviewer #2: The topic of this paper is well chosen, with clear research objects, research purpose, and research questions, possessing certain research value. The research methods are basically scientific and reasonable, and the research analysis and discussion are detailed and thorough. The research hypotheses are basically verified. The references are comprehensive, authoritative and academic.

The research subjects are relatively clear and representative. The article takes political speakers as an example to investigate the interaction between Role Congruity Theory (RCT) and Social Identity Theory (SIT) in the context of gender perception. The research question is relatively clear, and a hypothesis has been proposed: under the synergistic effect of RCT and SIT, male participants will show consistent preference for male speakers, while female participants will not show this bias. But what is the source of the participants mentioned in the article as “Ninety-five participants participating in this online survey”? What are the identities of participants? What is the rationale for the quantity or samples selection?

The article clearly states that it investigates the interaction between the RCT and SIT in the context of gender perception among political speakers. That is to say, it revolves around the speakers, but the evaluation is restricted to exclude writing lexemes.

In order to gain a more comprehensive understanding of the gender impact in political language processing, it is necessary to explore whether the results of current research can be replicated in oral and broader contexts. The sample size of the research object needs to be larger, and the empirical research tools are well researched, but the discourse corpus studied is not classified and not clear enough. The conclusion section is somewhat simple, and the authority of the references needs to be improved.

Reviewer #3: To begin, the paper hinges on constructs such as pejorative weight or left-right political connotation. Please define them in plain, operational terms. Briefly situate these constructs within established scholarly work on stance, indexicality, and evaluative meaning, and add the corresponding citations.

Second, explain why a continuous slider (VAS) was chosen for judgments that are often categorical or thresholded in everyday use. A short justification (comparability across items, sensitivity to subtle distinctions) and a note on trade-offs will strengthen credibility.

Additionally, enrich the sample description beyond gender and age by reporting recruitment channels, regional spread, education, and political interest so readers can see the social context in which these meanings are situated.

As the topic involves gender and politics, please explain who designed and selected the lexemes, what steps you took to avoid introducing your own ideological framing, and whether the instrument was piloted with politically diverse individuals.

Finally, you write that “the lexemes were carefully selected based on existing linguistic literature on the connotation of political terms on the left–right spectrum.” Please specify the literature you rely on and provide specific citations.

Overall, the topic is timely and the design is promising, but the paper would benefit from stronger construct definitions, reflexivity, context or transferability detail, and interpretive balance anchored in concrete literature and examples.

**Do you want your identity to be public for this peer review?** For information about this choice, including consent withdrawal, please see our Privacy Policy

Reviewer #1: **Yes:** Jing Hou

Reviewer #2: **Yes:** Xin Yan

Reviewer #3: No

---

## [Author Response · Author response to Decision Letter 1]

21 Dec 2025

Reviewer #1

We would like to thank Reviewer #1 for their comments. We believe their suggestions helped to substantially improve the quality of this work, including but not limited to overall clarity and readability, structural progression, the credibility of sample and item recruitment, and overall methodological soundness.

1. With the intent to promote readability and conciseness, we rephrased the example sentences mentioned by the reviewer. Furthermore, we actively engaged in improving reader comprehension throughout the rest of the manuscript.

2. To clarify in which way our study adds to the current corpus of research, we revised the summary part of our introduction, directly addressing literary gaps this research attempts to fill. (from line 135)

3. To clarify the conceptual and theoretical basis of the two evaluative dimensions used in the current study, pejorative weight and political connotation, we substantially restructured the methods section and introduced an entirely new subsection dedicated to these evaluative measures (see subsection ‘Pejorative weight and political connotation along a left-right spectrum’ from line 220).

4. We agree with the reviewer’s views that both the sample and lexeme recruitment passages lacked detailed information. We provided information on the recruitment means and inclusion criteria in the section ‘Participants and anonymity’ (from line 161). Further, we added an entirely new subsection ‘Lexeme pool’ (from line 188) to the methods section, explaining the lexeme curation process in more detail and adding crucial literature references we did not cover in the first version of the manuscript.

5. We agree with the reviewer on that the discussion would benefit from a clearer structure and more streamlined conceptual progression. Thus, we clearly restated our hypothesis and summarized key findings in the beginning of the discussion section, followed by a progression from a broader interpretation of the findings to a more detailed interpretation in the light of RCT and a subsequent turn to SIT. We closed the discussion by providing a summarizing interpretation involving the interactive effects of RCT and SIT. Finally, we informed future work about the implications of the interplay between RCT and SIT and gave recommendations on how to more appropriately disentangle the two.

Reviewer #2:

We would like to thank Reviewer #2 for their valuable comments. We believe their suggestions have significantly improved the quality of this work, enhancing its overall methodological soundness, clarity and comprehensiveness of the limitations and conclusion, as well as the authority of the references.

1. As pointed out by the reviewer, we enriched the sample description by additional characteristics, such as education level and mean SDO expression. Furthermore, we added information on the recruitment means as well as the rationale behind the sample size calculation (from line 161).

2. Limitations: We revised the limitations section (from line 494) of this paper by elaborating on the presentation modality of the lexemes: We elaborated on the shortcoming of exclusively using written lexemes and that future research is advised to not only replicate the mechanisms at play in the current study in a spoken language context, but also to situate political lexemes in a larger discourse context, such as statements or entire speeches.

3. Here we want to provide a response to the last paragraph of the reviewer’s comment:

a. We agree, to comprehensively draw generalizable conclusions about the presented gender mechanisms, one needs to investigate these in the context of oral and broader context frames. We elaborated on this argument and provided recommendations on how to collectively achieve generalizable results in the limitations section (from line 494).

b. We acknowledge the notion that our sample size needs to be larger and refer to the newly added part on the rationale of extracting a suitable sample size (from line 166).

c. We added significant information on the curation of the lexeme pool and provided corresponding literature references. We substantially re-structured the methods section and formed new subsections, one of which dedicated to the lexeme pool itself (from line 188).

d. To improve the conclusion section (from line 512), we summarized our results in more detail while maintaining a focus on clear language and readability.

e. The authority of the references has been improved as a consequence of the collective, valuable propositions and comments of all reviewers.

Reviewer #3:

We appreciate Reviewer #3 for their insightful feedback. We are confident that their recommendations have greatly enhanced the quality of this work, particularly in terms of the robustness of the methodology, the validity and scholarly basis of the evaluative measures, the clarity of construct definitions, and the credibility of the references.

1. We elaborated on the literary foundation of our two main evaluative dimensions, pejorative weight and political connotation along a left-right spectrum. We discussed their integration into and comparability with established measures, differential framings and criticism present in the literature, and reported relevant references. Secondly, we discussed the option of using VAS as an evaluative format considering relevant literature. For readability and clearer structural progression, a new subsection ‘Pejorative weight and political connotation along a left-right spectrum’ was created (from line 220).

2. We agree with the reviewer’s suggestion to report sample characteristics in more detail to provide information the social context in which this study was situated. We extended the ‘Participants and anonymity’ (from line 161) section by recruitment channels as well as educational and political information (represented by the average SDO scores).

3. Lexeme selection: We agree with the reviewer’s views that the curation/selection process needs to be described in more detail as the lexemes represent the core stimuli of the current study. A new subsection called ‘Lexeme pool’ (from line 188) has been added to the methods-section, elaborating on the specifics of the lexeme curation and selection process in more detail, while also adding crucial literature references we did not cover in the first version of the manuscript.

4. Literature regarding lexeme selection: In the course of describing the process of lexeme selection in more detail, we specified relevant literature in the newly added section ‘Lexeme pool’ (from line 188).

---

## [Editor Report · Decision Letter 1]

30 Jan 2026

Gender favoritism in derogatory and non-derogatory political discourse.

PONE-D-25-31394R1

Dear Dr. Hons,

We’re pleased to inform you that your manuscript has been judged scientifically suitable for publication and will be formally accepted for publication once it meets all outstanding technical requirements.

Kind regards,

Xiaoming Tian, Ph.D.

Academic Editor

PLOS One
---

## [Editor Report · Acceptance letter]

PONE-D-25-31394R1

PLOS One

Dear Dr. Hons,

I'm pleased to inform you that your manuscript has been deemed suitable for publication in PLOS One. Congratulations! Your manuscript is now being handed over to our production team.

Kind regards,

on behalf of

Dr. Xiaoming Tian

Academic Editor

PLOS One